# Frailty Assessment for Risk prediction in Gynecologic Oncology patients undergoing surgery and chemotherapy (FARGO) study protocol: Rationale and design of a multi-centre prospective cohort study

Julie My Van Nguyen[1]*, Danielle Vicus[2], Liat Hogen[2], Tiffany Zigras[3], Guillaume Paré[4,5,6], Michael Chong[4,5,6], Yetiani Roldan Benitez[7], P. J. Devereaux[7,8,9], Sandra Ofori[8,10], Flavia K. Borges[7,8,10], Emily Di Sante[10], Denise Miletic[10], Olivia Panus[10], Jessica Vincent[10], Chinthanie Ramasundarahettige[8,10], Sofia Nene[9], Ameen Patel[8], Maura Marcucci[7,10,11]

**1** Division of Gynecologic Oncology, Department of Obstetrics and Gynecology, McMaster University, Hamilton, Ontario, Canada, **2** Division of Gynecologic Oncology, Department of Obstetrics and Gynecology, University of Toronto, Toronto, Ontario, Canada, **3** Division of Gynecologic Oncology, Department of Obstetrics and Gynecology, University of Toronto, Mississauga, Ontario, Canada, **4** Population Health Research Institute, David Braley Cardiac, Vascular and Stroke Research Institute, Hamilton, Ontario, Canada, **5** Thrombosis and Atherosclerosis Research Institute, Hamilton, Ontario, Canada, **6** Department of Pathology and Molecular Medicine, McMaster University, Hamilton, Ontario, Canada, **7** Department of Health Research Methods, Evidence, and Impact, McMaster University, Hamilton, Ontario, Canada, **8** Department of Medicine, McMaster University, Hamilton, Ontario, Canada, **9** Division of Geriatrics, Department of Medicine, McGill University, St Mary's Hospital Centre, Montreal, Quebec, Canada, **10** Population Health Research Institute, McMaster University and Hamilton Health Sciences, Hamilton, Ontario, Canada, **11** Clinical Epidemiology and Research Centre, Humanitas University & IRCCS Humanitas Research Hospital, Milan, Italy

* nguyenjmv@hhsc.ca

## Abstract

### Background

There is considerable variability in how older adults with cancer tolerate and recover from surgery and systemic treatments. A greater understanding of individual trajectories is crucial in guiding personalized treatment decisions. Frailty may explain these inter-individual differences. Despite emerging evidence on the association between perioperative frailty assessment and outcomes after noncardiac surgery, there is limited data in gynecologic oncology. A perioperative cardiovascular risk assessment, recommended by scientific guidelines, is widely adopted in noncardiac surgery, often as the only standardized perioperative risk stratification approach. While based on robust evidence on the association with cardiovascular complications and overall mortality, it might be insufficient to predict other essential surgical, oncologic and patient-important outcomes.

**Data availability statement:** Data sharing is not applicable to this article as no datasets were generated or analyzed. Access to data will be restricted due to ethical concerns and available upon request after the completion of the study. The Population Health Research Institute (PHRI) believes the dissemination of clinical research results is vital and sharing of data is important. PHRI prioritizes access to data analyses to researchers who have worked on the study for a significant duration, have played substantial roles, and have participated in raising the funds to conduct the study. PHRI's balances the length of the research study, and the intellectual and financial investments that made it possible with the need to allow wider access to the data collected. Upon completion of the study, data will be disclosed only upon request and approval of the proposed use of the data by a Review Committee. Full details of our policy are available on request. Contact information@phri.ca.

**Funding:** JMVN: Hamilton Health Sciences (including the Early Career Research Award and the New Investigator Fund), the Juravinski Hospital and Cancer Centre Foundation, the Population Health Research Institute (Transforming Tomorrow Today grant) and the Hamilton Academic Health Sciences Organization (Innovation Fund) MM: Physicians' Services Incorporated (PSI) Foundation SO: career research award from McMaster University Department of Medicine. FKB: career research award by Hamilton Health Sciences The sponsors did not play a role in study design, data collection and analysis, decision to publish or preparation of the manuscript. URLs: https://www.hamiltonhealth.ca/, https://www.psifoundation.org/, https://research.mcmaster.ca/funding/hamilton-academic-health-sciences-organization-hahso-afp-innovation-grant-competition/, https://www.phri.ca/collaborations/transforming-tomorrow-today/.

**Competing interests:** DV- I have read the journal's policy and the authors of this manuscript have the following competing interests: Advisory board: GSK, Knight and Integra The other authors have declared that no competing interests exist.

## Methods

The FARGO study is a multi-centre prospective cohort study targeting 280 patients aged 55 or older undergoing surgery, with or without chemotherapy, for a suspected or confirmed gynecologic malignancy. The primary objective is to evaluate the predictive value of the Frailty Phenotype measured preoperatively, compared with the currently used perioperative risk assessment (cardiovascular risk assessment based on the Revised Cardiac Risk Index, age, and occurrence of myocardial injury after non-cardiac surgery) in predicting the composite outcome of all-cause death or new disability at six months after surgery. Secondary objectives include comparing the predictive value of the Frailty Phenotype with that of the Clinical Frailty Scale; evaluating the performance of a preoperative frailty assessments on other postoperative complications, chemotherapy tolerance, and 1-year recurrence-free survival; exploring the added predictive value of a dynamic perioperative frailty assessment repeated 28 days after surgery; assessing the acceptability of frailty assessments by physicians and patients; and establishing a biobank to investigate frailty biomarkers.

## Discussion

The findings could have important implications for risk stratification, planning and tailoring surgical and oncologic care for older adults with gynecologic malignancies. Our study emphasizes patient-centered outcomes and stakeholders' perspectives.
**Trial registration:** Clinicaltrials.gov Identifier: NCT05738252

## Introduction

### Background and rationale

The standard of care for gynecologic malignancies, such as ovarian, endometrial and cervical cancers, typically involves multimodal approaches, including surgery and systemic therapy. While the extent and distribution of the disease are primary factors in determining the treatment sequence, patient performance status also plays a critical role in decision-making. Decisions regarding surgery and chemotherapy, such as candidacy, sequencing, and regimen selection, significantly influence toxicity, morbidity, and survival outcomes [1,2]. Therefore, individualized assessments of patient prognosis and treatment-associated risks at several points along a patient's treatment trajectory are crucial. These tailored evaluations form the foundation for optimizing oncologic outcomes and quality of life, and minimizing complications.

Older patients often experience adverse outcomes compared to the younger and healthier patient populations typically included in oncology clinical trials that establish the standard of care [3–7]. They experience a greater risk of complications, long-term side effects impairing quality of life, and shorter survival [8–10]. This disparity is also evident in Gynecologic Oncology, where for example the morbidity and mortality rates of uterine cancer among older patients continue to increase despite

**Abbreviations:** 3D-CAM, 3-minute diagnostic confusion assessment method; AUC, area under the curve; CABG, coronary artery bypass graft; CAM, confusion assessment method; CAM-ICU, confusion assessment method for ICU patients; CFS, clinical frailty scale; CT Scan, computed tomography scan; DVT, deep venous thrombosis; ECG, electrocardiogram; FACT-G7, functional assessment of cancer therapy – general – 7 item version; FARGO, Frailty Assessment for Risk Prediction in Gynecologic Oncology patients undergoing surgery and chemotherapy study; FP, frailty phenotype; GCP, good clinical practice; GCS, Glasgow coma scale; hsTnT, high-sensitivity troponin T; ICH, international conference on harmonisation; IWRS, interactive web registration system; JHCC, Juravinski hospital and cancer centre; LBBB, left bundle branch block; MINS, myocardial injury after noncardiac surgery; MRI, magnetic resonance imaging; NACT, neoadjuvant chemotherapy; OADRI, Ottawa acceptability of decision rules instrument; OC, operations committee; OCREB, Ontario cancer research ethics board; OS, overall survival; PACU, post-anesthesia care unit; PCI, percutaneous coronary intervention; PE, pulmonary embolism; PHRI, population health research institute; PI, principal investigator; PROs, patient reported outcomes; QoL, quality of life; qSOFA, quick Sequential Organ Failure Assessment; RCRI, revised cardiac risk index; RDI, relative dose intensity; ROC, receiver operator characteristic; SC, steering committee; TnT, Troponin T; URL, upper reference limit; VISION, Vascular events in Noncardiac Surgery Patients Cohort Evaluation; WHODAS 2.0, World Health Organization Disability Assessment Schedule 2.0.

novel treatment options [11]. There remains considerable variability in how older adults with cancer tolerate and recover from surgery and systemic treatments [3–7]. As life expectancy increases and chronic disease management improves, chronological age and comorbidities alone fail to fully capture the risk of adverse outcomes or predict treatment responses in this population. Moreover, studies have shown that older patients with cancer often prioritize maintaining function and quality of life over extending survival [12].

Current risk-assessment tools often rely on predictors such as age and comorbidities or focus solely on short-term postoperative complications. They frequently overlook critical patient-centered outcomes such as quality of life, functional recovery, and long-term disability. These limitations underscore the pressing need for more comprehensive strategies and tools to assess risk, enabling personalized decision-making and treatment plans that better address the complexity and heterogeneity of older adults with cancer. Frailty could explain the inter-individual differences in health, function, and recovery, offering prognostic value beyond traditional clinical predictors such as age, comorbidities and cancer stage [13–17].

### Age-related frailty in the perioperative and oncological setting

Frailty describes a state of vulnerability, often age- and disease-related, resulting from diminished physiological reserves across multiple systems. It can be assessed using various methods, including questionnaires, physical tests, comprehensive clinical evaluations, or imaging [18–20]. Although no single frailty measure is considered the gold standard, the most widely studied and utilized definition is the Frailty Phenotype, developed by Fried et al. in the Cardiovascular Health Study and the Women's Health and Aging Studies [18,21,22]. This physical domain-focused model defines frailty as meeting at least three of five criteria: weak muscle strength, slow gait speed, unintentional weight loss, exhaustion, and low physical activity. Over time, additional frailty frameworks have been proposed, incorporating multiple domains beyond the physical to capture a broader spectrum of vulnerability [18–20].

In large cohort and population-based studies of noncardiac surgeries, preoperative frailty has been strongly associated with worse outcomes, including increased risks of short-term morbidity, falls, disability, higher resource utilization, long-term institutional placement, and 1-year mortality [23–27]. However, these studies largely exclude or minimally represent patients undergoing surgery for gynecologic malignancies, leaving a significant knowledge gap.

The prevalence of frailty in oncology may be as high as 50% [24]. In patients living with cancer, emerging evidence suggests that frailty may predict adverse oncologic outcomes [25,28,29], treatment tolerance, and all-cause mortality, independently of chronological age [8,17,24,30,31]. Among gynecologic oncology populations, studies have suggested that frailty is associated with adverse cancer-specific outcomes and treatment-related morbidity, such as reduced tolerance to chemotherapy, increased risk of treatment toxicity and perioperative complications, and decreased overall survival. However, the existing studies are small, retrospective, or focused on specific malignancies [32–39].

## Current perioperative risk assessment in gynecologic oncology

Currently, there are no gold-standard tools for perioperative risk stratification tailored specifically to gynecologic oncology patients. While older adults undergoing noncardiac surgery are at risk for various complications, much of the research has focused on cardiovascular events and their preoperative risk stratification. Cardiovascular complications are particularly common in older patients and represent a leading cause of postoperative mortality [40,41].

Guidelines from organizations such as the Canadian Cardiovascular Society and the European Society of Cardiology [42,43] provide clear, straightforward algorithms for perioperative cardiovascular risk assessment. These typically begin with evaluating patient medical history and calculating a risk score, most commonly using the Revised Cardiac Risk Index (RCRI). The RCRI includes six factors, each contributing one point: history of ischemic heart disease, cerebrovascular disease, congestive heart failure, preoperative insulin use, preoperative creatinine levels above 177 μmol/L, and high-risk surgery [44].

For patients with an RCRI score of 1 or higher, or for those aged 65 and older with an RCRI score of 0, the Canadian guidelines recommend postoperative troponin monitoring for the first three days after surgery [42]. This recommendation is based on evidence that myocardial injury after noncardiac surgery (MINS), defined as postoperative troponin elevation due to myocardial ischemia, is a strong independent predictor of 30-day mortality, regardless of symptoms or electrocardiogram (ECG) changes [45]. MINS accounts for the highest attributable risk (34%) of 30-day mortality among major postoperative complications [45] and is strongly associated with 1-year mortality [46,47]. Therefore, while MINS is categorized within cardiovascular risk, its occurrence is a significant predictor of overall postoperative outcomes.

Due to its strong prognostic value and the simplicity of algorithm implementation, cardiovascular risk stratification is widely adopted as the primary or sole formal perioperative risk assessment, including in gynecologic oncology. However, the landmark studies underpinning these recommendations included limited representation of gynecologic oncology patients and did not account for cancer-specific risk factors, such as cancer type and stage [44–46].

In contrast, geriatric multidimensional assessments, which evaluate a broader range of risks and vulnerabilities, are increasingly recommended in international oncology and perioperative guidelines for older adults before treatment initiation [47–50]. Despite these recommendations, such assessments are rarely integrated into routine practice, particularly in gynecologic oncology, where a focused cardiovascular approach remains dominant. This gap highlights the need for more comprehensive and tailored approaches to risk stratification in this patient population.

## Feasibility and acceptability of frailty assessment and role of biomarkers

Resource and time constraints are known barriers to routine adoption of more comprehensive perioperative risk evaluation, including frailty assessments [51–54]. For older or unwell gynecologic oncology patients, the additional time, physical tests, and detailed questionnaires required for such assessments could feel burdensome. This underscores the need to prioritize not only the predictive accuracy of frailty assessments but also their feasibility and acceptability in routine clinical practice. Research aimed at identifying and refining practical, patient-centered methods for frailty measurement is therefore essential.

Frailty biomarkers may offer a promising solution and could augment or even replace clinical assessments, enabling more streamlined and scalable evaluations. One such potential biomarker is mitochondrial DNA-copy number (mtDNA), measurable in peripheral blood leukocytes using a cost-effective and scalable assay. mtDNA-CN declines with age, and low mtDNA-CN is a marker of mitochondrial dysfunction, energy reserve depletion, oxidative stress, and systemic inflammation [55]. Mitochondrial dysfunction has been correlated with frailty and its adverse outcomes [55–58]; lower mtDNA-CN is associated with specific components of frailty assessment including with slower gait speed and weaker grip strength [59]. Another emerging biomarker is DNA methylation, which has been used to develop epigenetic frailty risk scores. DNA methylation clocks are the widely accepted instruments to measure biological aging.[60] DNA methylation, as opposed to the DNA sequence, which is overall preserved throughout the lifespan, is indeed the epigenetic process that

can capture changes in response to environmental and lifestyle exposures and thus yield greater amount of potentially prognostic information than chronological age. These scores have shown correlations with frailty both at baseline and during long-term follow-up, highlighting their potential as prognostic tools [61]. Such innovations could reduce the burden on patients and clinicians alike, facilitating the broader adoption of frailty assessments in gynecologic oncology and beyond.

### The need and role of the FARGO study

There is a clear need for robust prospective research to evaluate the role of frailty assessments through clinical assessment and biomarkers in gynecologic oncology patients who are surgical candidates. Understanding the added value of frailty assessments, either as a complement to or replacement for current cardiovascular-focused approaches, is critical.

In addition, research is needed to explore the feasibility and acceptability aspects of the adoption of frailty assessments in this population. To address these gaps, we designed the Frailty Assessment for Risk prediction in Gynecologic Oncology patients undergoing surgery and chemotherapy (FARGO) study, a multicentre prospective cohort study of patients aged 55 and older.

### Study objectives

In patients aged 55 or older with confirmed or suspected gynecologic cancer undergoing laparotomy surgery with or without chemotherapy:

### Primary objective

To evaluate the performance of preoperative frailty assessment based on the Frailty Phenotype compared to a perioperative cardiovascular risk assessment (combining the RCRI, age, and occurrence of MINS), in predicting the composite outcome of all-cause death or new disability at six months after surgery.

### Secondary objectives

1. To compare the predictive performance of *different preoperative/perioperative frailty assessments* for all-cause death or new disability at 6 months after surgery. The assessments will include:

   a. different frailty tools, such as the whole Frailty Phenotype, the single components of the Frailty Phenotype, and the Clinical Frailty Scale; and

   b. a *dynamic* perioperative frailty assessment (based on a frailty assessment repeated 28 days after surgery, in addition to a preoperative frailty assessment).

2. To explore the predictive performance of frailty assessments upon *chemotherapy-related outcomes*, including completion, total dose and number of cycles received, treatment toxicity and side effects, decisional regret, and impact on function.

3. To explore the value of frailty assessment *when added* to a perioperative cardiovascular risk assessment and age, with or without other clinical predictors, in predicting all-cause death or new disability at 6 months after surgery.

   To explore the predictive performance of a preoperative frailty assessment and of a preoperative cardiovascular risk assessment based on RCRI, age and MINS upon.

4. *postoperative outcomes at 28 days after surgery*, including all-cause death or new disability, major vascular events, infection and sepsis, bleeding, new clinically relevant atrial fibrillation/flutter, acute congestive heart failure, length of stay, unplanned admission to intensive care unit, and delirium.

5. all-cause death or new disability *at 1 year* after surgery.

6. *long-term postoperative outcomes*, including major vascular events at 6 months and 1 year after surgery, infection and sepsis at 6 months and 1 year after surgery, and all-cause death at 6 months and 1 year after surgery.

7. *oncologic outcomes* of progression-free survival and overall cancer-specific survival up to 1 year after surgery.

8. To measure *feasibility* and *acceptability* of frailty and disability assessments to patients and healthcare providers.

9. To establish a biobank to explore potential blood biomarkers of frailty.

## Methods

### Study design

This is a multicentre prospective cohort study of patients aged ≥55 years-old with confirmed or suspected gynecologic cancer who are candidates for laparotomy with or without chemotherapy.

### Eligibility criteria

**Inclusion criteria.** To be eligible, patients should meet both criteria 1 and 2:

1. Age 55 or older at registration.

2. Undergoing laparotomy for:

   a. cytoreduction of stage II-IV ovarian or endometrial/uterine cancer, with or without neoadjuvant chemotherapy.

   b. any stage endometrial, uterine or cervical cancer, where laparoscopy is deemed unfeasible or high-risk due to comorbidities.

   c. a pelvic mass deemed highly suspicious for malignancy.

   d. a gynecologic malignancy recurrence.

We included patients aged 55 and older as the relationship between age and outcome in oncological populations does not seem to be linear [17,62,63]. Treatment tolerance and oncologic outcomes might start declining at a relatively younger age than what is traditionally considered older [14,64,65]: and frailty may help explain these differences.

**Exclusion criteria.** Eligible patients would not be included if they meet any of the following exclusion criteria:

1. Unable to provide informed consent.

2. Require urgent surgery within 24 hours of first consultation with the gynecologic oncology team.

3. Undergoing neoadjuvant radiation therapy.

4. Previously documented history of dementia.

5. Cognitive, language, vision, or hearing impairment impacting the ability to understand the directions for completing the study instruments.

6. Participating in a clinical trial investigating a new neoadjuvant systemic therapy.

### Study flow and study patient groups

Fig 1 presents the timeline of enrolment, assessments and visits. Fig 2 presents the FARGO patient flow diagram. We expect that some of the eligible patients (group A) will undergo surgery without neoadjuvant chemotherapy. Group A will

| | Before surgery | | | After surgery | | | |
| | Patients Not Receiving Chemotherapy | Patients Receiving Chemotherapy | | All Patients | | | |
| | Preoperative Assessment[1] | Before chemotherapy[2,3] | After chemotherapy[2,4] | Day 0 to +3 (+14 days) | 28-Day Visit[5] | 6-Month Visit[5] | 1-Year Visit[5] |
|---|---|---|---|---|---|---|---|
| Patient Consent and Enrolment | X | X | | | | | |
| Baseline Sociodemographic and Medical History | X[5] | X[5] | | | | | |
| Frailty Phenotype | X | X | X | | X | | |
| Clinical Frailty Scale | X | X | X | | X | | |
| Troponin Measurement | | | | X | | | |
| Disability Assessment (WHODAS 2.0) | X[5] | X[5] | X[5] | | X | X | X |
| Delirium Assessment (3D-CAM) | | | | X[6] | | | |
| Health-Related Quality of Life Questionnaire (FACT-G7) | X | X | | | X | X | X |
| Clinical Outcomes | | | | X | X | X | X |
| Chemotherapy-related outcomes | | | X | | | X | |
| Blood Drawing for Biobanking | X | X | X | | X | X | X |

[1] Completed within 45 days before surgery

[2] Refers to the round of chemotherapy closest to the surgery date

[3] Completed within 45 days before chemotherapy

[4] Completed at least 18 days from the last infusion before surgery and within 45 days before surgery

[5] Can be conducted in-person or by telephone/videoconference

[6] Conducted twice a day

**Fig 1. Study timeline: Enrolment, assessments, visits and data collection.**

thus undergo only one preoperative study visit (baseline) within 45 days before their surgery. Patients undergoing neoadjuvant chemotherapy (group B) will have a first baseline study assessment within 45 days before their first chemotherapy treatment. In this group, a second preoperative study visit will be repeated after neoadjuvant chemotherapy is considered completed, at least 18 days post-chemotherapy cycle and within 45 days before surgery. After completion of neoadjuvant chemotherapy, a small proportion of patients will no longer be deemed eligible for surgery by the treating physician (group C). Group C will be included in the evaluation of chemotherapy-related outcomes and will also be asked to complete a follow-up visit 6 months from the date of their registration. Group D, not represented in Fig 2, will consist of those who will eventually not undergo any surgery or neoadjuvant chemotherapy, after being deemed initially eligible and having completed the baseline assessment. We expect group D to consist of an even smaller proportion of recruited patients. For Group D, a follow-up visit at 6 months from the baseline visit will be completed. Group A and B will complete the study

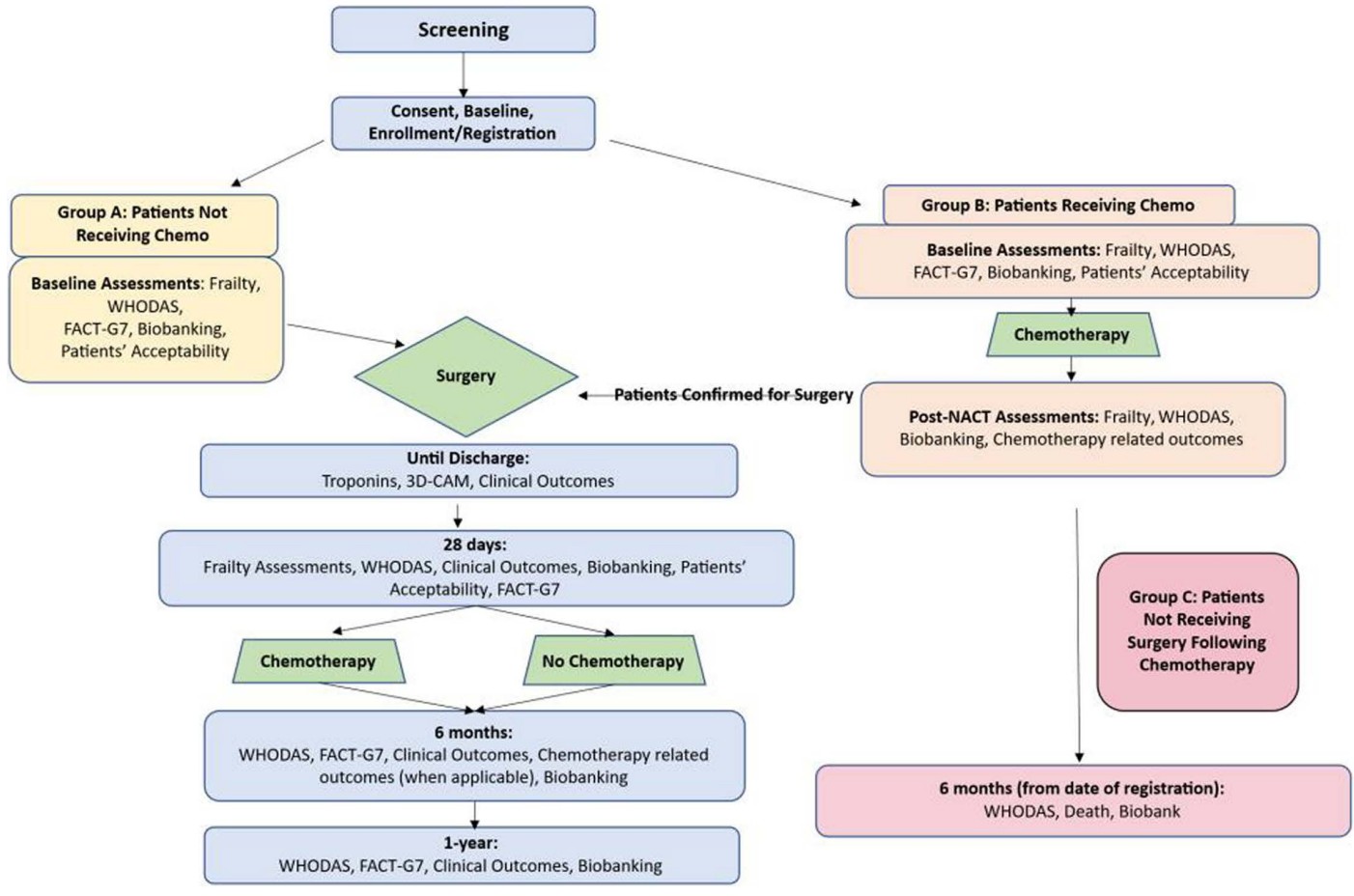

**Fig 2. Study Flow Diagram.**

assessments and follow-ups. Some patients from group A and group B will undergo adjuvant chemotherapy following surgery. This will not affect their study timeline.

Clinical follow-up at each participating centre is based on international oncology guidelines such as those of the National Comprehensive Cancer Network for follow-up for each primary disease site.

## Definitions and measurements

### Frailty assessments

The primary frailty measurement will be based on the Frailty Phenotype. Research personnel (including research assistants) will assess the following 5 criteria: weak muscle strength (decreased grip strength measured with a dynamometer), slow gait speed (measured on the 15-ft or 4-m walking test), unintentional weight loss (> 10 lbs in the last 12 months), exhaustion (self-reported), and low physical activity (low weekly energy expenditure, calculated using a modified version of the Adult Compendium of Physical Activities [66,67]). For the primary purpose of the study, we will measure the patient's current level of physical activity; with an exploratory intent, we will gather further information on the usual activity level (before the diagnosis of cancer was made, or before the cancer likely started affecting their performance and lifestyle). Participants receive 1 point for each criterion met, for a total score of: 0–1, not frail (robust); 2–3, intermediate frail (pre-frail); 4–5, frail.

Frailty will also be assessed using the Clinical Frailty Scale (CFS) [22]. The CFS is a global frailty scale according to which the patient is assigned a category from 1 ("Very fit") to 9 ("Terminally ill"), as a summative clinical judgment; frailty is present if a category≥4 is assigned.

Research personnel are trained to assess the CFS before administering the Frailty Phenotype so that they judgement on the CFS is not affected by the results of the Frailty Phenotype assessment. Both frailty assessments are performed at baseline and at 28 days after surgery. Patients undergoing neoadjuvant chemotherapy will have two preoperative frailty assessments, at enrolment before neoadjuvant chemotherapy, and after the completion of neoadjuvant chemotherapy.

### Study outcomes

The predictive performance of perioperative frailty and cardiovascular risk assessments will be assessed upon the following clinical outcomes.

**Primary clinical outcomes.** The primary outcome is all-cause death or new disability at 6 months after surgery. We will measure disability in several domains (including cognition, mobility, self-care, participation) using the 12-item World Health Organization Disability Assessment Schedule (WHODAS 2.0) [68], a patient-reported outcome tool widely validated in surgical older populations [69]. New disability will be defined as a disability score≥25% at follow-up, or a score increase of ≥ 8% for those already disabled at baseline. These thresholds were selected based on previous validation studies [68,70], where a threshold of 8% difference in WHODAS score was found to be clinically meaningful based on normative data. Disability measured by WHODAS has been found to be of high patient-importance in surveys of older surgical patients [71] and is used for other surgical oncology populations [72] and populations receiving chemotherapy [73].

**Secondary clinical outcomes.** Secondary outcomes include:

- All cause death or new disability at 28 days and at 1 year after surgery,

- All cause death at 28 days, 6 months, and 1 year after surgery,

- Oncologic outcomes, including progression-free survival and cancer-specific death, up to 1 year after surgery,

- Major vascular complications (at 28 days, 6 months, and 1 year after surgery) defined as a composite of vascular death, and non-fatal myocardial infarction (or myocardial injury for the 28-day time point), stroke, symptomatic proximal venous thromboembolism, and cardiac arrest,

- Bleeding Independently Associated with Mortality after noncardiac Surgery (BIMS) at 28 days after surgery,

- New clinically important atrial fibrillation at 28 days, 6 months, and 1 year after surgery,

- Acute congestive heart failure at 28 days, 6 months, and 1 year after surgery,

- Infection, and infection with sepsis (at 28 days, 6 months, and 1 year after surgery),

- Unplanned admission to the intensive care unit (ICU) during the index hospital admission for surgery,

- Length of stay (during the index hospital admission for surgery),

- In-hospital delirium (during the index hospital admission for surgery),

- Chemotherapy-related outcomes, which include: tolerance as objectively defined by total dose received and time to completion, patient's decisional regret (i.e., distress or remorse after a health care decision, such as to undergo chemotherapy), change in health-related function (as measured by the WHODAS 2.0), and change in health-related quality of life (as measured by the Functional Assessment of Cancer Therapy – General – 7 Item Version [FACT-G7]).

Appendix I includes the study clinical outcome definitions.

**Acceptability outcomes.** We will collect time and rate of completion of the FP and CFS (and sub-items of the FP). Acceptability of frailty assessment according to physicians will be defined based on ease of use and clinical relevance, according to relevant items of the Ottawa Acceptability of Decision Rules Instrument (OADRI) [74]. Patient's acceptability of frailty and disability assessments will be measured administering a brief exit questionnaire.

**Measurement and adjudication of study outcomes.** To reduce the influence from the knowledge of the participants' frailty status (predictor) on the evaluation of disability (outcome), whenever possible the WHODAS 2.0 will be administered by different study personnel from those who administered the frailty assessments. In this manner, they will also be blinded to the patients' RCRI and postoperative troponin level.

Study outcomes will be adjudicated according to a prespecified adjudication plan that defines processes to adjudicate various outcomes, including adjudication by validation and algorithm coding, by verification, and evaluation by adjudicators. The adjudicators consist of physicians with expertise in perioperative outcomes who will be blinded to patients' frailty assessments, RCRI and troponin status of the study participants. We will use the decisions of the adjudication process for all statistical analyses.

## Patient recruitment and informed consent

At the participating sites at academic hospitals in Canada, research personnel will screen the list of patients aged 55 years or older, diagnosed with a suspected or confirmed gynecologic malignancy and requiring a) a surgical intervention performed by laparotomy or b) NACT, with plan for interval surgery via laparotomy, who are scheduled for any preoperative or pre-chemotherapy visit at the participating centres. Clinics involved in the preoperative or pre-chemotherapy assessment of the potential study population as per local practice (such as gynecologic oncology clinics, internal medicine clinics, anesthesiology clinics, etc.) will be engaged in the patient screening process. If a patient meets the criteria, study personnel will approach the patient to obtain their informed consent. Based upon the clinic schedule and access to the patients, a combination of remote and written consent may be used based upon the patient preferences. If the patient chooses remote consent, the signed informed consent form will be returned to the site prior to initiation of study procedures. These informed consent options may be provided to reduce burden on the patient population. Patient recruitment commenced on June 12th, 2023 and is expected to be completed by December 31, 2025. The list of study sites can be obtained by contacting the corresponding author.

## Data collection and follow-up

Fig 1 presents the timeline of the study data collection.

**Preoperative and in-hospital data collection.** Preoperatively, research personnel will collect information on sociodemographics and medical history. This information will be used to calculate the patient RCRI. Participants will also complete the frailty assessments, the Functional Assessment of Cancer Therapy – General – 7 Item Version (FACT-G7), and the World Health Organization Disability Assessment Schedule 2.0 (WHODAS 2.0). Patients undergoing NACT before surgery will undergo these assessments twice: once before, and once following completion of their NACT, before surgery.

In hospital, all patients will have troponin measurement on postoperative days 1, 2 and 3. Study personnel will administer the 3-minute diagnostic interview for the Confusion Assessment Method (3D-CAM) twice daily, on the first 3 days after surgery. Blood collection for the optional biobank component will be completed before surgery, before and after NACT, at the 28-day post-surgery visit, 6-month post-surgery visit, and 1-year post-surgery visit. Data collection is expected to be completed by March 31, 2027 and results are expected by May 31, 2027.

**Follow-Up.** Research personnel will follow participants while in hospital and collect data regarding clinical events. Follow-up visits will be conducted by study personnel (either by telephone, videoconference or in person) at 28 days, 6 months, and 1-year post-surgery. Efforts will be made to conduct these visits in concomitance with clinical follow-ups.

Frailty assessments will be repeated at the 28-day post-operative visit. The WHODAS 2.0 and FACT-G7 will be administered at the 28-day, 6-month and 1-year visits. Data on clinical events and chemotherapy-related outcomes will be collected at each study visit.

**Evaluation of the acceptability of frailty assessments.** Acceptability of frailty assessment according to physicians will be measured using the Ottawa Acceptability of Decision Rules Instrument (OADRI). Patient's acceptability of frailty and disability assessments will be evaluated in a sub-set of patients by qualitatively administering an exit interview. Appendix II provides further details.

**Blood sample collection and biobank.** Only participants who provide a specific separate consent will be enrolled in the biobank study. We will collect blood samples from patients at baseline (before and after the NACT and/or before surgery), and subsequently at 28 days after surgery, 6 months and 12 months. We will collect EDTA tubes to obtain both plasma and whole blood for DNA analysis. Samples will be shipped to the Clinical Research Laboratory and Biobank – Genetic and Molecular Epidemiology Laboratory (CRLB-GMEL), Hamilton, Ontario, and stored at –160°C in liquid nitrogen vapour.

mtDNA-CN will be measured using a plasmid-normalized quantitative PCR assay which concurrently amplifies segments of the tRNA leucine mitochondrial gene and the beta-2 microglobulin nuclear gene to quantify the relative number of mitochondrial to nuclear copies [75]. Cellular and cell-free mtDNA-CN will be assayed separately. For measurement of cell-free mtDNA-CN, a specialized lysis protocol, "MitoQuicLy" (2023) will be applied to plasma samples [76].

DNA methylation will be measured using the CRLB-GMEL lab's targeted and custom methylation array. This was designed from systematically querying epigenetic databases (epigenome-wide association study (EWAS) catalogue & datahub) [77,78] and the literature for large (N > 1000) EWAS. Based on this search, we selected ~50,000 CpG sites shown in the literature to be highly accurate (Pearson's correlation coefficient >0.9) epigenetic predictors. [60,79,80]

## Sample size

The FARGO study is sized for the primary objective. We expect a 20% rate of the primary outcome of death or new disability at 6 months after surgery, based on previous literature on patients undergoing major abdominal surgery [81,82]. We performed preliminary analyses of the VISION [83] cohort. Based on these analyses, we expect that, in our study population, the ability of the baseline RCRI and age combined with the occurrence of MINS to predict 6-month death or new disability, measured as Area Under the receiver operating characteristic Curve (AUC), will not be greater than 0.600 (range 0.550–0.600, Appendix I). A sample size of 250 patients will allow at least 80% power to detect a difference of at least 0.130 in the AUC for a model including only the FP-frailty assessment, compared with the AUC for the model including preoperative RCRI and age combined with MINS (i.e., AUC for FP-frailty model of at least 0.680) [23]. The table in the Appendix III displays the sample size calculation based on these assumptions for a range of scenarios, based on the comparison of two Receiving Operating Characteristic (ROC) curves, 2-sided test, alpha = 0.05, and assumed correlation between the 2 predictive models = 0.5.

We will include 280 patients to conservatively account for ≤ 10% of participants who will be lost before or during surgery, and in whom troponin measurement to assess for MINS will not be possible. In fact, we expect that after recruitment, significantly fewer patients will not undergo or survive surgery. Based on our eligibility criteria, we expect that approximately 50% of participants will undergo chemotherapy, before and/or after surgery, and will be included in our exploratory secondary analyses on chemotherapy outcomes.

## Statistical analyses

The primary analysis will compare the performance of preoperative frailty assessment based on the FP and perioperative cardiovascular risk assessment (combining the preoperative RCRI, age and occurrence of MINS), in predicting the composite of all-cause death or new disability at 6 months after surgery. We will compare the predictive performance of FP

and of RCRI, age and MINS with the primary outcome by comparing the AUC calculated using logistic regression models. We will perform one logistic regression model including frailty as predictor, and one logistic regression model including the baseline RCRI, age and MINS as predictors. For each model we will then calculate the c-statistics (corresponding to the AUC) and compare them using a non-parametric approach [84]. We will include in the primary analysis only study participants who have at least one postoperative troponin measurement. We anticipate that this will exclude from the primary analysis a very small proportion of patients initially enrolled in the study (i.e., ≤ 2%). We will then perform a sensitivity analysis, accounting for missing troponin data and reasons why troponin was not measured (early death or other reasons). We will secondarily include the surgery indication, based on our inclusion criteria, in both models, as covariate and as effect modifier.

As part of our secondary objectives, we will also evaluate the utility of frailty in predicting 6-month disability or death, *when added* to the RCRI, age and MINS as predictors by calculating the net absolute reclassification improvement [85].

To compare the predictive performance of different frailty assessments, we will use a similar approach as for the primary outcome: we will compare the c-statistics of logistic regression models, each including one type of frailty assessment. To examine the added value of a dynamic perioperative frailty assessment on the primary outcome, we will use a logistic regression model and add preoperative frailty and change in frailty at 28 days after surgery, either as absolute change or as percent change as covariates.

Logistic regression or time-to-event models will be performed for the association of frailty and perioperative cardiovascular risk assessment with other outcomes, including chemotherapy-related outcomes, as appropriate.

Effects sizes will be presented together with 95% confidence intervals. The threshold for statistical significance will be set as $p < 0.05$.

The physicians' and patients' acceptability scores will be compared between frailty instruments using Wilcoxon signed rank tests. The mean and standard deviation (in seconds) required to complete each frailty instrument will be compared using Student t tests. The numbers of missing values will be compared using 2x2 tests.

## Ethical considerations

This study will be conducted in compliance with the protocol, principles laid down in the Declaration of Helsinki, and all applicable laws and regulations. Before study initiation, each site must have written and dated approval/favorable opinion from the Institutional Review Board/Independent Ethics Committee for the protocol and consent form. The FARGO study was granted Ethics approval by the Ontario Cancer Research Ethics Board (OCREB), Clinical Trials Ontario Project identification number 3714. Amendments to the protocol have received Ethics approval.

## Current status

To date 177 participants have been recruited across three Canadian sites. We expect recruitment to be completed by the fall of 2025.

## Discussion

The multicentre prospective cohort FARGO study will add new knowledge to inform and potentially improve risk assessment and individualized care for older adults undergoing surgery and systemic therapy for gynecologic malignancies. Ultimately, our results could help anticipate the impact of cancer and its treatments on patients, their families, and healthcare resources, potentially improving healthcare delivery. By involving stakeholders in evaluating the feasibility and acceptability of frailty assessments, the study will provide guidance for sustainable, innovative, and patient-centered improvements in clinical practice.

To our knowledge, our study will be the first to compare frailty assessments to current standard perioperative risk evaluations in predicting outcomes and treatment tolerance in gynecologic oncology. Additionally, it is novel in measuring frailty dynamically throughout a patient's treatment trajectory, and in contributing new knowledge on the role of frailty

biomarkers. Furthermore, while primarily focused on frailty as a predictor, the study will also address critical gaps in understanding perioperative cardiovascular risks and outcomes in a population previously underrepresented in large epidemiological studies of cardiovascular complications.

Given the significant underrepresentation of older adults in clinical trials that establish standard treatments [3–7], research tailored to this complex population is essential. The FARGO study seeks to support the development of cancer-specific, evidence-based guidelines for personalized risk stratification and management. This will ensure older patients' unique vulnerabilities are addressed, prioritizing not only survival but also functional recovery and quality of life.

## Appendices

### Appendix I. Clinical outcomes definitions

The following list provides the definitions for the clinical outcomes for which the predictive ability of perioperative frailty and cardiovascular risk assessments will be assessed in the study.

**New disability (at 28 days, 6 months, and 1 year after surgery).** Disability status is determined using the 12-item World Health Organization Disability Assessment Schedule (WHODAS 2.0). New disability at any time point is based according to the following criteria that account for baseline disability scores:

1. For individuals with a disability score of <25% at baseline, new disability is defined as a disability score ≥25% at follow-up;

2. For individuals with a disability score ≥25%, new disability is defined as an increase in disability score of ≥8% at follow-up.

Definition of new disability at 28 days, 6 months, and 1 year after surgery will use as baseline disability score the WHODAS 2.0 score measured before surgery.

**Sub-classification of death.** Vascular death is defined as any death with a vascular cause and includes those deaths following a myocardial infarction, cardiac arrest, stroke, cardiac revascularization procedure (i.e., percutaneous coronary intervention [PCI] or coronary artery bypass graft [CABG] surgery), pulmonary embolus, hemorrhage, or deaths due to an unknown cause. Non-vascular death is defined as any death due to a clearly documented non-vascular cause (e.g., trauma, infection, malignancy).

**Oncologic outcomes (at 6 and 12 months).** They will include:

1. Progression-free survival, defined as the time from treatment initiation to tumor progression or recurrence or death from any cause, or to the date of censoring at the last time the subject was known to be alive.

2. Cancer-specific death is death directly attributable to the primary gynecologic cancer or directly related to its treatment, in the absence of other causes of death.

For this study and the evaluation of frailty predictive performance, treatment initiation will be first defined as the time of surgery, and the evaluated frailty assessment will be the one performed before surgery (even in patients who had NACT). Secondarily, treatment initiation will be defined as initiation of any cancer-specific treatment, whether NACT or surgery, and the evaluated frailty assessment will be the one performed before the first cancer-specific treatment, whether NACT or surgery. Cancer progression/recurrence will be defined as a measurable progression/recurrence documented on imaging.

**Chemotherapy-related outcomes.** They will include the following, measured at the end of any chemotherapy treatment (whether neo-adjuvant or adjuvant) when considered terminated:

1. Total dose received, defined as Relative Dose Intensity (RDI), as calculated as the percentage of the standard dose that was administered, using the formula below:

$$\text{Carboplatin RDI (\%) = total dose administered/ total standard dose } * 100$$

$$\text{Paclitaxel RDI (\%) = total dose administered/ total standard dose } * 100$$

2. Time to completion of all chemotherapy cycles, expressed as number of days.

3. Patient's decisional regret, defined as "distress or remorse after a (health care) decision," assessed using the Decisional Regret scale, a validated a 5-item scale [86].

4. Change in health-related function or well-being, defined as the difference in WHODAS 2.0 score after chemotherapy compared to before chemotherapy.

5. Change in health-related quality of life, defined as the difference in FACT-G7 score after chemotherapy compared to before chemotherapy.

**Major vascular complications (at 28 days, 6 months, and 1 year after surgery).** This is defined as a composite of vascular death, and non-fatal myocardial infarction (or myocardial injury for the 28-day time point), stroke, symptomatic proximal venous thromboembolism, and cardiac arrest. The definitions for the components of the composite are the following:

**Myocardial Injury after Noncardiac Surgery (MINS).** MINS is defined as any myocardial infarction (as defined below), and any elevated troponin (higher than the local laboratory threshold) judged to be due to myocardial ischemia (i.e., without evidence of a non-ischemic etiology [e.g., chronic elevation, PE, sepsis, cardioversion]) that occurred within the first 28 days after the day of surgery. The only exceptions to the definition of an elevated troponin will be to use a higher threshold for TnT of ≥30 ng/L, and for hsTnT of 20 to <65 ng/L with an absolute change of at least 5 ng/L or an hsTnT level ≥65 ng/L. These thresholds for TnT and hsTnT are based upon data from a large international prospective perioperative cohort study that established troponin thresholds that were independently associated with 30-day mortality after noncardiac surgery [44].

**Myocardial infarction.** If the diagnostic criteria for myocardial infarction includes an elevated troponin, then the definition of MINS must be met to fulfill the diagnostic criteria for myocardial infarction (4th universal definition). The diagnosis of myocardial infarction requires any one of the following criteria:

1. Detection of a rise or fall of a cardiac biomarker (preferably troponin) with at least one value above the 99th percentile of the upper reference limit (URL) together with evidence of myocardial ischemia with at least one of the following:

   A. ischemic signs or symptoms (i.e., chest, arm, neck, or jaw discomfort; shortness of breath, pulmonary edema),

   B. development of pathologic Q waves present in any two contiguous leads that are ≥ 30 milliseconds,

   C. new or presumed ECG changes indicative of ischemia (i.e., ST segment elevation [≥ 2 mm in leads $V_1$, $V_2$, or $V_3$ OR ≥ 1 mm in the other leads], ST segment depression [≥ 1 mm], or symmetric inversion of T waves ≥ 1 mm) in at least two contiguous leads,

   D. new left bundle branch block (LBBB),

   E. new cardiac wall motion abnormality on echocardiography or new fixed defect on radionuclide imaging, or

   F. identification of intracoronary thrombus on angiography or autopsy

2. Cardiac death, with symptoms suggestive of myocardial ischemia and presumed new ischemic ECG changes or new LBBB, but death occurred before cardiac biomarkers were obtained, or before cardiac biomarker values would be increased.

3. PCI-related myocardial infarction is defined by elevation of a troponin value (>5 x 99th percentile URL) in patients with a normal baseline troponin value (≤99th percentile URL) or a rise of a troponin measurement >20% if the baseline values are elevated and are stable or falling. In addition, either (i) symptoms suggestive of myocardial ischemia or (ii) new ischemic ECG changes or (iii) angiographic findings consistent with a procedural complication or (iv) imaging demonstration of new loss of viable myocardium or new regional wall motion abnormality, are required.

4. Stent thrombosis associated with myocardial infarction when detected by coronary angiography or autopsy in the setting of myocardial ischemia and with a rise and/or fall of cardiac biomarker values with at least one of value above the 99th percentile URL.

5. CABG-related myocardial infarction is defined by elevation of cardiac biomarker values (>10 x 99th percentile URL) in patients with a normal baseline troponin value (≤99th percentile URL). In addition, either (i) new pathological Q waves or new LBBB, or (ii) angiographic documented new graft or new native coronary artery occlusion, or (iii) imaging evidence of new loss of viable myocardium or new regional wall motion abnormality.

6. For patients who are believed to have suffered a myocardial infarction within 28 days of a MINS event or within 28 days of a prior myocardial infarction, the following criterion for myocardial infarction is required:

Detection of a rise or fall of a cardiac biomarker (preferably troponin) with at least one value above the 99th percentile of the URL and 20% higher than the last troponin measurement related to the preceding event together with evidence of myocardial ischemia with at least one of the following:

 A. ischemic signs or symptoms (i.e., chest, arm, neck, or jaw discomfort; shortness of breath, pulmonary edema),

 B. development of pathologic Q waves present in any two contiguous leads that are ≥ 30 milliseconds,

 C. new or presumed new ECG changes indicative of ischemia (i.e., ST segment elevation [≥ 2 mm in leads $V_1$, $V_2$, or $V_3$ OR ≥ 1 mm in the other leads], ST segment depression [≥ 1 mm], or symmetric inversion of T waves ≥ 1 mm) in at least two contiguous leads,

 D. new LBBB, or

 E. new cardiac wall motion abnormality on echocardiography or new fixed defect on radionuclide imaging,

 F. identification of intracoronary thrombus on angiography or autopsy

**Stroke.** Stroke is defined as a new focal neurological deficit thought to be vascular in origin with signs or symptoms lasting more than 24 hours or leading to death.

**Symptomatic Proximal Venous Thromboembolism.** Venous thromboembolism that includes symptomatic PE or symptomatic proximal deep vein thrombosis.

**Symptomatic Pulmonary Embolism (PE).** The diagnosis of symptomatic PE requires symptoms (e.g., dyspnea, pleuritic chest pain) or signs (e.g., hypoxia, increased work of breathing) and any one of the following:

1. A high probability ventilation/perfusion lung scan,

2. An intraluminal filling defect of segmental or larger artery on a helical CT scan,

3. An intraluminal filling defect on pulmonary angiography, or

4. A positive diagnostic test for DVT (e.g., positive compression ultrasound) and one of the following:

 A. non-diagnostic (i.e., low or intermediate probability) ventilation/perfusion lung scan, or

 B. non-diagnostic (i.e., subsegmental defects or technically inadequate study) helical CT scan.

**Symptomatic Proximal Deep Venous Thrombosis (DVT).** The diagnosis of symptomatic proximal DVT requires:

1. symptoms or signs that suggest DVT (e.g., leg pain or swelling),

2. thrombosis involving the popliteal vein or more proximal veins for leg DVT OR axillary or more proximal veins for arm DVTs.

   Any of the following defines evidence of vein thrombosis:
   A. a persistent intraluminal filling defect on contrast venography (including on computed tomography),

   B. noncompressibility of one or more venous segments on B mode compression ultrasonography, or

   C. A clearly defined intraluminal filling defect on doppler imaging in a vein that cannot have compressibility assessed (e.g., iliac, inferior vena cava, subclavian).

**Nonfatal cardiac arrest.** Nonfatal cardiac arrest is defined as successful resuscitation from either documented or presumed ventricular fibrillation, sustained ventricular tachycardia, asystole, or pulseless electrical activity requiring cardiopulmonary resuscitation, pharmacological therapy, or cardiac defibrillation.

**Infection, and infection with sepsis (28 days, 6 months, and 1 year).** Infection is defined as a pathologic process caused by the invasion of normally sterile tissue or fluid or body cavity by pathogenic or potentially pathogenic organisms.

The Third International Consensus Definitions Task Force defines sepsis as a "life-threatening organ dysfunction due to a dysregulated host response to infection." Based on the Third International Consensus Definitions for Sepsis and Septic Shock (Sepsis-3) criteria, sepsis will require a quick Sequential Organ Failure Assessment (qSOFA) Score ≥2 points due to infection. The qSOFA includes the following items and scoring system:

1. Altered mental status (1 point),

2. systolic blood pressure of 100 mm Hg or less (1 point), and

3. respiratory rate of 22 breaths/min or more (1 point).

**In-hospital delirium.** Delirium during the first 3 days after surgery or before discharge from the hospital, based on Confusion Avssessment Method (CAM). According to CAM, patients are diagnosed with delirium if they meet the first 2 criteria (acute onset with fluctuating course, AND attention deficit), and at least one of the second 2 criteria (disorganized thinking OR altered level of consciousness). Participants will be screened for postoperative delirium while in hospital, twice daily, during the first 3 days after surgery or until discharge (if before 3 days), by research personnel, using the 3D-CAM, or the CAM-ICU any time the participants are in the PACU or in ICU.

**Bleeding Independently Associated with Mortality after noncardiac Surgery (BIMS).** BIMS is a bleeding event meeting any of the following 3 criteria:

1. Leading to a postoperative hemoglobin <70 g/L,

2. Requiring transfusion of one or more units of red blood cells,

3. Judged to be the immediate cause of death.

**New clinically important atrial fibrillation.** The definition of new clinically important atrial fibrillation requires the documentation of atrial fibrillation or atrial flutter of any duration on an ECG or rhythm strip, which results in angina, congestive heart failure, symptomatic hypotension, or requires treatment with a rate controlling drug, antiarrhythmic drug, or electrical cardioversion.

**Acute congestive heart failure.** The definition of congestive heart failure requires at least one of the following clinical signs (i.e., any of the following signs: elevated jugular venous pressure, respiratory rales/crackles, crepitations, or presence of S3) and at least one of the following:

1. Radiographic findings (i.e., vascular redistribution, interstitial pulmonary edema, or frank alveolar pulmonary edema), or

2. Heart failure treatment implemented with diuretics with documented clinical improvement.

**Appendix II – Measuring acceptability of frailty assessments**

**Physicians' acceptability.** Study personnel will invite a convenience sample of consenting physicians involved in the perioperative clinical assessment of the study population (e.g., the Gynecologic Oncology group, the Perioperative Medicine Service, etc.), with no experience with frailty tools, to participate in the administration of the FP and CFS. The study personnel will subsequently administer the acceptability questionnaire (OADRI) to the physicians. The questionnaire is a modified version of the OADRI where only items deemed relevant are retained. Physicians are asked to rate each item, for each tool, along a 7-point Likert scale including the following options: strongly disagree, moderately disagree, slightly disagree, no opinion/don't know, slightly agree, moderately agree, strongly agree.

| | **Score on Likert-scale**<br>Strongly disagree, moderately disagree, slightly disagree, no opinion/don't know, slightly agree, moderately agree, strongly agree. |
|---|---|
| The tool is easy to use. | |
| The tool is useful in my practice. | |
| The wording of the tool is clear and unambiguous. | |
| My colleagues support use of the tool. | |
| Patients benefit from use of the tool. | |
| Using the tool results in improved use of resources. | |
| The evidence supporting the tool is flawed. | |
| I'm already using another tool or similar strategy. | |
| The tool does not account for an important clinical cue. | |
| The environment I work in makes it difficult to use the tool. | |

**Patients' acceptability.** Patient's acceptability of frailty assessments will be evaluated administering a questionnaire in a purposeful sample of the main study participants. The first 10 patients at each site will be asked about the acceptability of the Frailty Phenotype Assessment and WHODAS questionnaire shortly following their completion at the baseline study visit. The next 10 patients (different from those asked at baseline) will be asked about the acceptability of the Frailty Phenotype and WHODAS questionnaire shortly following their completion at the 28-day follow-up visit. In total, 20 patients at each site will be asked about the acceptability of the FP and WHODAS, 10 at each indicated visit.

| **1**. Was the patient asked about the acceptability of the Frailty Phenotype Assessment? | No<br>Yes |
|---|---|
| 2. On a scale of 0–5, with 0 being not easy at all, and 5 being the easiest, how easy was it to complete the Frailty Phenotype assessment? | 0<br>1<br>2<br>3<br>4<br>5<br>No opinion<br>Don't know |

| 3. The Frailty Phenotype Assessment took a long time to complete | Strongly disagree<br>Disagree<br>Neither Disagree or Agree<br>Agree<br>Strongly Agree |
|---|---|
| 4. The language used in the Frailty Phenotype Assessment was easy to understand | Strongly disagree<br>Disagree<br>Neither Disagree or Agree<br>Agree<br>Strongly Agree |
| 5. One or more tasks of the Frailty Phenotype Assessment was/were physically or psychologically/emotionally burdensome to complete | Strongly disagree<br>Disagree<br>Neither Disagree or Agree<br>Agree<br>Strongly Agree |
| 6. On a scale of 0–5, with 0 being not helpful at all, and 5 being the most helpful, how helpful for understanding your health was it to complete the Frailty Phenotype assessment? | 0<br>1<br>2<br>3<br>4<br>5<br>No Opinion<br>Don't Know |
| 7. Would you complete the Frailty Phenotype assessment again? | Yes<br>No |
| 1. Was the patient asked about the acceptability of WHODAS? | No<br>Yes |
| 2. On a scale of 0–5, with 0 being not easy at all, and 5 being the easiest, how easy was it to complete the WHODAS questionnaire? | 0<br>1<br>2<br>3<br>4<br>5<br>No opinion<br>Don't know |
| 3. The WHODAS Assessment took a long time to complete | Strongly disagree<br>Disagree<br>Neither Disagree or Agree<br>Agree<br>Strongly Agree |
| 4. The language used in the WHODAS assessment was easy to understand | Strongly disagree<br>Disagree<br>Neither Disagree or Agree<br>Agree<br>Strongly Agree |
| 5. One or more items of the WHODAS questionnaires was/were psychologically/emotionally burdensome to complete | Strongly disagree<br>Disagree<br>Neither Disagree or Agree<br>Agree<br>Strongly Agree |
| 6. On a scale of 0–5, with 0 being not helpful at all, and 5 being the most helpful, how helpful for understanding your health was it to complete the WHODAS questionnaire? | 0<br>1<br>2<br>3<br>4<br>5<br>No Opinion<br>Don't Know |

| 7. Would you complete the WHODAS question- naire again? | Yes No |
|---|---|

## Appendix III. Sample size calculations

The sample size calculation is based on the comparison of two receiver operator characteristic curves (ROC), 2-sided test, with alpha = 0.05. The assumed AUC for the RCRI, age, and MINS model is 0.55 (AUC1). The assumed AUC of the frailty assessment model is between 0.65–0.75 (AUC2). The assumed AUC difference is approximately 0.13. We expect an incidence of the primary outcome (all-cause death or disability at 6 months after surgery) to be 20%.

A sample size of 250 patients will give us at least 80% of power to detect a difference of at least 0.130 in the AUC for a model including solely the FP-frailty assessment, compared with the AUC for the model including preoperative RCRI and age combined with MINS.

The assumed correlation between the two predictive models is 0.50 for the positive group (those experiencing the outcome) and 0.50 for the negative group (those not experiencing the outcome).

We will include 280 patients to conservatively account for ≤10% who will be lost before or during surgery, or in whom troponin measurement will not be possible.

Software: PASS 13 (NCSS, LLC. Kaysville, Utah, USA. www.ncss.com)

**Sample size calculation.**

| AUC1 | AUC difference | AUC2 | Target power | Sample size |
|---|---|---|---|---|
| 0.55 | 0.10 | 0.65 | 80% | 405 |
| 0.55 | 0.10 | 0.65 | 85% | 465 |
| 0.55 | 0.10 | 0.65 | 90% | 545 |
| 0.55 | 0.13 | 0.68 | 80% | 240 |
| 0.55 | 0.13 | 0.68 | 85% | 280 |
| 0.55 | 0.13 | 0.68 | 90% | 325 |
| 0.55 | 0.15 | 0.70 | 80% | 185 |
| 0.55 | 0.15 | 0.70 | 85% | 210 |
| 0.55 | 0.15 | 0.70 | 90% | 245 |
| 0.60 | 0.10 | 0.70 | 80% | 425 |
| 0.60 | 0.10 | 0.70 | 85% | 485 |
| 0.60 | 0.10 | 0.70 | 90% | 570 |
| 0.60 | 0.13 | 0.73 | 80% | 250 |
| 0.60 | 0.13 | 0.73 | 85% | 290 |
| 0.60 | 0.13 | 0.73 | 90% | 340 |
| 0.60 | 0.15 | 0.75 | 80% | 190 |
| 0.60 | 0.15 | 0.75 | 85% | 220 |
| 0.60 | 0.15 | 0.75 | 90% | 255 |

## Supporting information

**S1 File. FARGO PROTOCOL v1.0_2022-08-10.**
(PDF)

**S2 File. FARGO_PROTOCOL_Final_v5.0_2025-04-25.**
(PDF)

**S3 File. SPIRIT FARGO v2 2025-01-06.**
(DOC)

## Author contributions

**Conceptualization:** Julie My Van Nguyen, Danielle Vicus, Liat Hogen, Tiffany Zigras, Guillaume Pare, Michael Chong, Maura Marcucci.

**Data curation:** Julie My Van Nguyen, Danielle Vicus, Liat Hogen, Tiffany Zigras, Guillaume Pare, Michael Chong, Yetiani Roldan Benitez, Sofia Nene, Maura Marcucci.

**Funding acquisition:** Julie My Van Nguyen, Maura Marcucci.

**Investigation:** Julie My Van Nguyen, Danielle Vicus, Liat Hogen, Tiffany Zigras, Guillaume Pare, Michael Chong, Yetiani Roldan Benitez, P. J. Devereaux, Sandra Ofori, Flavia K. Borges, Emily Di Sante, Denise Miletic, Olivia Panus, Jessica Vincent, Chinthanie Ramasundarahettige, Sofia Nene, Ameen Patel, Maura Marcucci.

**Methodology:** Julie My Van Nguyen, Emily Di Sante, Denise Miletic, Olivia Panus, Jessica Vincent, Chinthanie Ramasundarahettige, Maura Marcucci.

**Project administration:** Julie My Van Nguyen, Maura Marcucci.

**Writing – original draft:** Julie My Van Nguyen, Maura Marcucci.

**Writing – review & editing:** Danielle Vicus, Liat Hogen, Tiffany Zigras, Guillaume Pare, Michael Chong, Yetiani Roldan Benitez, P. J. Devereaux, Sandra Ofori, Flavia K. Borges, Emily Di Sante, Denise Miletic, Olivia Panus, Jessica Vincent, Chinthanie Ramasundarahettige, Sofia Nene, Ameen Patel.

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
