## [Decision Letter · Decision Letter 0]

Dear Dr. Nguyen,

Thank you for submitting your manuscript to PLOS ONE. After careful consideration, we feel that it has merit but does not fully meet PLOS ONE’s publication criteria as it currently stands. Therefore, we invite you to submit a revised version of the manuscript that addresses the points raised during the review process.

We look forward to receiving your revised manuscript.

Kind regards,

Fabrizio D'Acapito, Ph.D,M.D.

Academic Editor

PLOS ONE

Journal Requirements:

Additional Editor Comments:

I thank the authors for submitting their study. The topic is very interesting from a research point of view and very relevant from a clinical point of view. I consider the study to be well constructed. I think the reviewers have made more than enough suggestions for further improvement of the paper, so I would just add one small comment: it might be useful to investigate how the criteria currently used (ASA, CCI,...) to assess patients match the proposed scores or could be integrated with them.

Reviewers' comments:

Reviewer's Responses to Questions

**Comments to the Author**

1. Does the manuscript provide a valid rationale for the proposed study, with clearly identified and justified research questions?

Reviewer #1: Yes

Reviewer #2: Yes

2. Is the protocol technically sound and planned in a manner that will lead to a meaningful outcome and allow testing the stated hypotheses?

Reviewer #1: Yes

Reviewer #2: Partly

3. Is the methodology feasible and described in sufficient detail to allow the work to be replicable?

Reviewer #1: Yes

Reviewer #2: Yes

4. Have the authors described where all data underlying the findings will be made available when the study is complete?

Reviewer #1: Yes

Reviewer #2: No

5. Is the manuscript presented in an intelligible fashion and written in standard English?

Reviewer #1: Yes

Reviewer #2: Yes

You may also provide optional suggestions and comments to authors that they might find helpful in planning their study.

Reviewer #1: Interesting study which will add to the current knowledge on assessment of the elderly patient with a gynecologic oncology disease.

Especially interesting is the addition of the combination of cardiac assessment and the available geriatric assessments and the incorporation of collecting material for biomarkers.

I do wonder whether drawing blood for 3 days postop to test for troponine might be to invasive and costly to incorporate in routine daily practice.

In clinical practice, the nature of the surgery (to include staging procedures or not) will be determined on the estimation whether a patient will be able to undergo chemotherapy and/or radiotherapy (especially in case of endometrial carcinoma). Could this study predict whether an elderly patient could tolerate the adjuvant therapy?

Reviewer #2: The study faced an interesting topic, frailty in patients with advanced gynecologic cancer. The protocol is detailed and with a robust background, statistical analysis is well conducted for the correct sample size of the study. However, some questions arise.

The primary objective of the study is to evaluate the predictive value of preoperative frailty phenotype in comparison with commonly used perioperative risk assessment tools such as the Revised Cardiac Risk Index in predicting the composite outcome of all-cause death or new disability at six months after surgery. I ‘m surprised that the authors completely omit in their evaluation commonly used preoperative assessment methods (ASA score, Charlson Index for example). Why didn’t the authors stratify patients with these methods? Renal failure, lung disease and so on are all factors that are commonly considered as risk factors for death and disability after surgery.

Regarding the secondary objective (the predictive value of the Frailty Phenotype with that of the Clinical Frailty Scale), the methods for measuring postoperative complications and chemotherapy tolerance is wrong and at all not normally used in the clinical practice. Why did the authors include cardiac morbidity only? Are lung and renal morbidity not relevant? Are hematological adverse events not relevant? If the authors want to grade post-operative outcomes, I suggest using for postoperative outcome the Clavien-Dindo classification and for chemotherapy adverse event the Common Terminology Criteria for Adverse Events (CTCAE), which include all the detrimental effects of chemotherapy.

Minor comments

The term “research personnel” is extremely vague. Who is going to measure frailty? The oncologist, the surgeons? A research nurse?

If the authors want to evaluate the oncological outcome, they should state how they want to organize the follow-up. I suggest just cite the national oncological guidelines follow-up schedule for each tumor histology.

Regarding the biobank part of the study, the methods of DNA analysis is lacking. The rationale should also better discussed.

**Do you want your identity to be public for this peer review?** For information about this choice, including consent withdrawal, please see our Privacy Policy

Reviewer #1: No

Reviewer #2: No

---

## [Author Response · Author response to Decision Letter 1]

14 Apr 2025

Dear PLOS One Editorial Board Members,

On behalf of my co-authors, I sincerely thank you for reviewing our manuscript titled: “Frailty Assessment for Risk prediction in Gynecologic Oncology patients undergoing surgery and chemotherapy (FARGO) study protocol: rationale and design of a multi-centre prospective cohort study”. Please see below a detailed response to your feedback. We are very grateful for your consideration.

Academic Editor comment:

I thank the authors for submitting their study. The topic is very interesting from a research point of view and very relevant from a clinical point of view. I consider the study to be well constructed. I think the reviewers have made more than enough suggestions for further improvement of the paper, so I would just add one small comment: it might be useful to investigate how the criteria currently used (ASA, CCI,...) to assess patients match the proposed scores or could be integrated with them.

Author response: Many thanks for the appreciation of our work.

The approved protocol does not explicitly state the suggested analyses, but we are collecting variables for ASA and comorbidity scores to include them as other clinical predictors in the multivariable analyses for one of the secondary objectives: “To explore the value of frailty assessment when added to a perioperative cardiovascular risk assessment and age, with or without other clinical predictors, in predicting all-cause death or new disability at 6 months after surgery.” To address this objective, we will indeed look at how frailty scores correlate with other clinical predictors and whether frailty scores add to the predictive accuracy of these other clinical predictors.

Reviewer #1:

Reviewer #1, Comment 1: Interesting study which will add to the current knowledge on assessment of the elderly patient with a gynecologic oncology disease. Especially interesting is the addition of the combination of cardiac assessment and the available geriatric assessments and the incorporation of collecting material for biomarkers.

Author response: Thank you for this feedback!

Reviewer #1, Question 1: I do wonder whether drawing blood for 3 days postop to test for troponin might be to invasive and costly to incorporate in routine daily practice.

Author response: Thank you for this question. This is standard practice based on guidelines such as those from the Canadian Cardiovascular Society and the European Society of Cardiology, as detailed from lines 131-143. Patients routinely get daily bloodwork after surgery (to include CBC, creatinine, electrolytes) and based on the above guidelines, troponins are also routinely added.

Reviewer #1, Question 2: In clinical practice, the nature of the surgery (to include staging procedures or not) will be determined on the estimation whether a patient will be able to undergo chemotherapy and/or radiotherapy (especially in case of endometrial carcinoma). Could this study predict whether an elderly patient could tolerate the adjuvant therapy?

Author response: Thank you for this question! Yes one of our secondary outcomes is to assess frailty and its association with tolerance to chemotherapy, “as objectively defined by total dose received and time to completion, patient’s decisional regret (i.e., distress or remorse after a health care decision, such as to undergo chemotherapy), change in health-related function (as measured by the WHODAS 2.0), and change in health-related quality of life (as measured by the Functional Assessment of Cancer Therapy – General – 7 Item Version [FACT-G7]).” (lines 337-342).

Reviewer #2:

Reviewer #2, Comment 1: The study faced an interesting topic, frailty in patients with advanced gynecologic cancer. The protocol is detailed and with a robust background, statistical analysis is well conducted for the correct sample size of the study.

Author response: Thank you for this feedback!

Reviewer #2, Question 1: The primary objective of the study is to evaluate the predictive value of preoperative frailty phenotype in comparison with commonly used perioperative risk assessment tools such as the Revised Cardiac Risk Index in predicting the composite outcome of all-cause death or new disability at six months after surgery. I ‘m surprised that the authors completely omit in their evaluation commonly used preoperative assessment methods (ASA score, Charlson Index for example). Why didn’t the authors stratify patients with these methods? Renal failure, lung disease and so on are all factors that are commonly considered as risk factors for death and disability after surgery.

Author response: Thank you for this question. The RCRI does account for several comorbidities: history of ischemic heart disease, cerebrovascular disease, congestive heart failure, preoperative insulin use, preoperative creatinine levels above 177 µmol/L (lines 135-137). Logistic regression analyses will include other important prognostic factors, such as ASA score and Charlson comorbidity index. The approved protocol does not explicitly state the suggested analyses, but we are collecting variables for ASA and comorbidity scores to include them as other clinical predictors in the multivariable analyses for our primary and secondary outcomes (as stated in one of our secondary objectives- “To explore the value of frailty assessment when added to a perioperative cardiovascular risk assessment and age, with or without other clinical predictors, in predicting all-cause death or new disability at 6 months after surgery.”)

Reviewer #2, Question 2: Regarding the secondary objective (the predictive value of the Frailty Phenotype with that of the Clinical Frailty Scale), the methods for measuring postoperative complications and chemotherapy tolerance is wrong and at all not normally used in the clinical practice. Why did the authors include cardiac morbidity only? Are lung and renal morbidity not relevant? Are hematological adverse events not relevant? If the authors want to grade post-operative outcomes, I suggest using for postoperative outcome the Clavien-Dindo classification and for chemotherapy adverse event the Common Terminology Criteria for Adverse Events (CTCAE), which include all the detrimental effects of chemotherapy.

Author response: Thank you for this feedback. As explained in our protocol and manuscript, the reason we used the cardiovascular risk assessment as a comparator is that this is the current clinical practice at our centres and, based on our discussions with colleagues in Canada and internationally, is the type of preoperative risk assessment most commonly implemented across centres. Hence, among the complications, we put emphasis on postoperative cardiovascular complications, which is what cardiovascular indexes are primarily expected to predict. This will be also an important contribution to our knowledge per se, since in the existing literature there is scanty data regarding the performance of these risk scores in the gynecologic oncologic population. With regards to postoperative and chemotherapy-related complications, we are indeed collecting all postoperative clinical events (including cardiovascular outcomes, nonvascular postoperative outcomes such as infections, pulmonary and renal events) and chemotherapy adverse events among the secondary outcomes. The main focus is on global outcomes such as mortality, disability, quality of life and chemotherapy tolerance, as this is what we expect frailty to help predict, rather than any specific complication.

Minor comments

Reviewer #2, Question 3: The term “research personnel” is extremely vague. Who is going to measure frailty? The oncologist, the surgeons? A research nurse?

Author response: It will be the research assistants (added to line 285).

Reviewer #2, Question 4: If the authors want to evaluate the oncological outcome, they should state how they want to organize the follow-up. I suggest just cite the national oncological guidelines follow-up schedule for each tumor histology.

Author response: Thank you, we have explained the follow-up schedule based on research visits, and we confirm that the follow-up schedule will be according to national oncological guidelines for each tumor histology. This was added to lines 279-281.

Reviewer #2, Question 5: Regarding the biobank part of the study, the methods of DNA analysis is lacking. The rationale should also better discussed.

Author response: Thank you, we have added 2 paragraphs on methods for mtDNA-CN analysis and DNA methylation analysis (lines 411-421). We have also added to the rationale (lines 174-179).

We sincerely thank you for your consideration of our manuscript, and we look forward to your feedback.

Julie Nguyen

---

## [Editor Report · Decision Letter 1]

Frailty Assessment for Risk prediction in Gynecologic Oncology patients undergoing surgery and chemotherapy (FARGO) study protocol: rationale and design of a multi-centre prospective cohort study

PONE-D-24-58425R1

Dear Dr. Nguyen,

We’re pleased to inform you that your manuscript has been judged scientifically suitable for publication and will be formally accepted for publication once it meets all outstanding technical requirements.

Kind regards,

Fabrizio D'Acapito, Ph.D,M.D.

Academic Editor

PLOS ONE

Additional Editor Comments (optional):

I have read with interest the replies provided to the reviewers and find them pointed and appropriate. I believe that the additional details added to the text have answered the questions raised by the reviewers.

I congratulate the authors for their work and hope to see the results of the FARGO Study soon.
---

## [Editor Report · Acceptance letter]

PONE-D-24-58425R1

PLOS ONE

Dear Dr. Nguyen,

I'm pleased to inform you that your manuscript has been deemed suitable for publication in PLOS ONE. Congratulations! Your manuscript is now being handed over to our production team.

Kind regards,

on behalf of

Dr. Fabrizio D'Acapito

Academic Editor

PLOS ONE